# Virulence Traits of Inpatient *Campylobacter jejuni* Isolates, and a Transcriptomic Approach to Identify Potential Genes Maintaining Intracellular Survival

**DOI:** 10.3390/microorganisms8040531

**Published:** 2020-04-07

**Authors:** Judit K. Kovács, Alysia Cox, Bettina Schweitzer, Gergely Maróti, Tamás Kovács, Hajnalka Fenyvesi, Levente Emődy, György Schneider

**Affiliations:** 1Department of Medical Microbiology and Immunology, University of Pécs Medical School, 7624 Pécs, Hungary; Kovacs.judit2@pte.hu (J.K.K.); sbetti0101@gmail.com (B.S.); fenyvesi.hajnalka@pte.hu (H.F.); levente.emody@aok.pte.hu (L.E.); 2Department of Biotechnology, Nanophagetherapy Center, Enviroinvest Corporation, 7632 Pécs, Hungary; Alysia.Cox@cshs.org (A.C.); kovacst@enviroinvest.hu (T.K.); 3Institute of Plant Biology, Biological Research Center, 6726 Szeged, Hungary; marotig@baygen.hu; 4Veterinary Medical Research Institute, Hungarian Academy of Sciences, 1581 Budapest, Hungary

**Keywords:** *Campylobacter jejuni*, clinical isolates, genomic approach, virulence potential, intracellular survival, transcriptomic analysis

## Abstract

There are still major gaps in our understanding of the bacterial factors that influence the outcomes of human *Campylobacter jejuni* infection. The aim of this study was to compare the virulence-associated features of 192 human *C. jejuni* strains isolated from hospitalized patients with diarrhoea (150/192, 78.1%), bloody diarrhoea (23/192, 11.9%), gastroenteritis (3/192, 1.6%), ulcerative colitis (3/192, 1.5%), and stomach ache (2/192, 1.0%). Traits were analysed with genotypic and phenotypic methods, including PCR and extracellular matrix protein (ECMP) binding, adhesion, and invasion capacities. Results were studied alongside patient symptoms, but no distinct links with them could be determined. Since the capacity of *C. jejuni* to invade host epithelial cells is one of its most enigmatic attributes, a high throughput transcriptomic analysis was performed in the third hour of internalization with a *C. jejuni* strain originally isolated from bloody diarrhoea. Characteristic groups of genes were significantly upregulated, outlining a survival strategy of internalized *C. jejuni* comprising genes related (1) to oxidative stress; (2) to a protective sheath formed by the capsule, LOS, N-, and O- glycosylation systems; (3) to dynamic metabolic activity supported by different translocases and the membrane-integrated component of the flagellar apparatus; and (4) to hitherto unknown genes.

## 1. Introduction 

*Campylobacter jejuni* is regarded as a major cause of diarrhoeal diseases in both industrialised and developing countries, mostly associated with the consumption of undercooked poultry meat and contaminated surface water [1]. *Campylobacter* infections are increasing around the world [2] with an annual average of 17 diagnosed cases per 100,000 people [3]. The clinical spectrum of campylobacteriosis ranges from asymptomatic to severe symptoms, such as bloody diarrhoea, sometimes embodied in post-infection sequelae, including Miller–Fisher and Guillain–Barré syndromes [4,5].

To reduce and prevent *Campylobacter* infections in humans, it is crucial to understand those virulence factors and molecular mechanisms that directly contribute to pathogenesis. There are still major gaps in our understanding, although the roles of major virulence properties, including its chemotaxis, motility, spiral shape, adhesion, and invasion abilities, are known. 

Numerous studies have documented the prevalence of specific coding regions in pathogenicity-associated features in human and animal isolates and examined correlations between their presence in the bacterial genome and disease severity [1,6,7,8,9], with controversial results. 

After the early elucidation of the pivotal role of flagella in pathogenesis [10], systemic research established that motility is based on a complex flagellar system built up by a membrane complex (Class I or Early genes), a Hook protein (Class II or Middle genes), and a filament (Class III or Late genes) [11]. The “corkscrew-like” motility [12,13] across the mucus membranes is facilitated by the spiral shape that is a unique feature of campylobacters and helicobacters, and is strictly regulated, leading to alterations in cell morphology of *C. jejuni* [14].

Adhesion is one of the most important factors contributing to colonization. *C. jejuni* binds first to mucin, then to some extracellular matrix proteins (ECMPs), including fibronectin and laminin, and finally directly onto the surface of epithelial cells. This process is facilitated by several factors, such as CadF, Peb1, Peb2, Peb3, Peb4, CapA, CjaA, FlpA, FbpA, JlpA, DocA, and other new candidates [15,16,17,18,19,20]. It is evident that adhesion is a very complex process that is influenced by the variability of several proteins, lipooligosaccharides (LOS), capsules (CPS) [21], and the unique *O*- or *N*-linked glycosylation systems [22,23,24,25]. Based on gene content and organization of the LOS biosynthesis loci, the LOS classes A, B, and C have been described [26]. Along with class-specific glycosyltransferases, these classes possess *neuBCA* genes that are required for sialic acid biosynthesis and a *cstII* gene that encodes a sialic acid transferase [26,27,28].

After successful colonization most *C. jejuni* strains have the capacity to invade upper epithelial cells and have been shown to enter gut tissue cells in vivo and in vitro [29,30,31,32,33]. Previous observations suggested that the invasion ability of an isolate could correlate with inflammation and the severity of the clinical outcome [30]. In vitro cell culture methods, based on INT407, Caco, HeLa, and T84 cell lines, have expanded our knowledge about the roles of specific cellular components in the invasion process [15,19,34,35,36,37,38]. The importance of certain flagellar genes (*flaA, flaB, flgB, flgE*, and *flaC*) during invasion is now known [39]. Although *C. jejuni* does not have a classical type III secretion system (T3SS) [40], the flagellar apparatus fulfils this role [41] and its presence is required for maximal cell invasion [39,42], and is also involved in the secretion and delivery of certain invasion antigens (CiaC and CiaD) into the cytoplasm of epithelial cells [19,37,43]. Along with the flagellar genes, the role of bacterial ABC transporter component PEB1 and an autotransporter protein CapA can mediate both adherence and invasion in host epithelial cells [16,18]. The vast majority of research has focused on the invasion of *C. jejuni* from the apical membrane, although some studies also demonstrated the importance of the basolateral route [34], where not only fibronectin, but laminin and collagen IV as well are present as potential adhesion targets [44].

Although *Campylobacter* produces several cytotoxins, only the cytolethal distending toxin (CDT) has been studied in detail [45]. CDT induces cell cycle arrest and interleukin 8 (IL-8) secretion from intestinal epithelial cells in culture by causing direct DNA damage, leading to progressive fragmentation of the nucleus [46,47,48,49,50].

The factors that maintain survival of the internalized *C. jejuni* in eukaryotic cells are still unclear. In contrast to other invasive bacteria, such as *Listeria monocytogenes*, *Shigella flexneri*, and *Salmonella typhimurium*, internalization of *C. jejuni* into epithelial cells occurs by a trigger mechanism [33] in a microtubule-dependent and actin-independent manner [51]. After internalization *C. jejuni* resides within a membrane-bound compartment that does not fuse with lysosomes [52]. Unique metabolic features, including the utilization of amino acids and organic acids as sole carbon sources [53] and application of a highly branched electron transport chain, allow *C. jejuni* to respire not only oxygen, but also other alternative electron acceptors [54,55]. How these processes functionally maintain the intracellular lifecycle, and what molecular occurrences support this stage of infection, remain elusive [55].

One of the key differences between infection of humans and chickens by *C. jejuni* is the apparently increased number of bacteria invading epithelial cells in the human host [56]. This suggests that both bacterial adherence to and entrance into epithelial cells may be critical steps that are essential for disease development [33].

Our aims with this study were (i) to assess the presence of known virulence-associated factors in *C. jejuni* strains isolated from hospitalized patients with a wide variety of symptoms in South-West Hungary; (ii) to reveal and compare the capacities of the isolates for ECMP binding, adhesion, and invasion; and (iii) to evaluate how *C. jejuni* remains internalized by using whole transcriptomic analysis (WTA). Deeper knowledge about the correlation of virulence attributes with clinical outcomes, and about the genes expressed in the internalized *C. jejuni* cell, could reveal novel mechanisms and facilitate the development of innovative control strategies.

## 2. Materials and Methods

### 2.1. Bacterial Strains and Growth Conditions

Altogether 192 *C. jejuni* strains were studied. A total of 190 *C. jejuni* isolates were collected from human diarrheal and non-diarrheal stool samples by using the routine isolation and identification method [57] in the Department of Bacteriology of the South Transdanubian Regional Public Health Institute. Repeat isolates that were isolated from patients excreting other bacterial pathogens, had known co-morbidity, received previous antibiotic therapy, or that were hospitalised for more than 24 h at the collection of samples, as identified from the laboratory request forms, were excluded from the study. Briefly, stool cultures were tested for *Salmonella, Shigella, Yersinia*, and thermophilic *Campylobacter* by standard culture methods. Colony morphology, oxidase positivity, Gram staining followed by microscopic morphology, and hydrolysis of hippurate and indoxyl acetate were applied to identify C. *jejuni*. Strains were conserved at −80 °C in tryptic soy broth (TSB, Scharlab, Barcelona Spain) containing 20% glycerol. Two wild-type reference strains (81–176 and NCTC 81116) were used as controls. Isolates were grown on Charcoal Cefoperazone Deoxycholate Agar (CCDA) at 42 °C under microaerophilic conditions generated by the MACS–MICS jar-system (Don Whitley Scientific, Shipley, UK). The jar system used the carbogen gas mixture (90% N_2_ and 10% CO_2_) to create microaerophilic conditions. Bacteria were grown for 24 h prior to all experiments. Colonies were collected, suspended, and their optical densities (OD) were synchronized to OD_600_ = 1 in phosphate-buffered saline (PBS). Semisolid (0.3%) brain–heart infusion (BHI) agar (Oxoid, Hampshire, UK) was used for motility tests. Strain CjTD-119 underwent transcriptional analysis in the third hour of internalization. This strain was one of seven strains isolated from a patient with bloody diarrhoea and showed high adhesion and high invasion. Before the study all 190 isolates were identified as individual strains based on their *SmaI* macro restriction patterns performed by Pulsed Field Gel Electrophoresis (PFGE) (https://www.cdc.gov/pulsenet/pdf/campylobacter-pfge-protocol-508c).

### 2.2. Polymerase Chain Reaction (PCR)

After 24 h growth on CCDA, bacterial cell counts were standardized by adjusting the OD_600_ to 1.0 (~4 × 10^8^ CFU/ mL). Total DNA was prepared by boiling 1 mL bacterial suspension for 10 min. The tested genes, primers, and annealing temperatures are listed in online Appendix A. PCR was performed in a DNA Thermal Cycler (Eppendorf, Hamburg, Germany) using standardized amplification parameters: 95 °C for 1 min for initial denaturation followed by 30 cycles of denaturation at 95 °C for 30 s, various annealing temperatures for 2 min, and an elongation step at 72 °C for 2 min. DNA bands were obtained with electrophoresis on 1% agarose gel, stained with ethidium bromide, and visualized in the BioCapt Imaging System (BioRad, Hercules, California, USA).

### 2.3. Solid-Phase Extracellular Matrix Protein (ECMP) Binding Assay

The solid-phase ECMP binding assay was performed as previously described [58] with slight modifications. The following ECMPs were applied: fibronectin from human foreskin fibroblast (Sigma-Aldrich, St. Louis, Missouri, USA), laminin (from Engelbreth–Holm–Swarm murine sarcoma basement membrane, Sigma-Aldrich), and collagen type IV (from human placenta, Sigma-Aldrich). Each well of the 96-well plates (Sarstedt, Nuembrecht, Germany) were filled with 100 µL aliquots of 10 µg/mL fibronectin, laminin, and collagen type IV in PBS, respectively, followed by overnight incubation at 4 °C. The next day wells were washed three times with 200 µL PBST (PBS, 0.5% 10 ×Tween 20), and blocked with 100 µL 2% bovine serum albumin (BSA, Sigma-Aldrich) for 2 h at room temperature. BSA was removed by washing the plates three times in PBST. Bacterial cells were grown as defined above, harvested in PBS, and adjusted to OD_600_ = 1.0. To each well 100 µL of the bacterial cell suspension was added and incubated at 37 °C for 3 h. Plates were washed three times with PBST, filled with 1 mL 1% PBS-Triton X-100 (Sigma-Aldrich), and incubated at 37 °C for 10 min to detach bound bacteria. The resuspended protein–bacterium complex (10 μL) was dropped and bled on CCDA. After 48 h incubation under microaerophilic conditions at 42 °C, colony forming units (CFUs) were counted, and the ratio of protein-binding ability was calculated. *Pseudomonas aeruginosa* was used as a positive control, binding collagen type IV with 0.16%, fibronectin with 0.16%, and laminin with 0.1%.

### 2.4. INT 407 Binding and Internalization Assay

Adhesion and invasion analysis of the clinical strains was performed on semi-confluent monolayer ATCC, INT 407 human embryonic intestine (jejunum and ileum) cell line (ATCC), on 24-well culture plates. Semi-confluent cell monolayers were prepared (3 × 10^5^ cells per well) in RPMI 1640 medium (BioWhittaker, Lonza, Basel, Switzerland) supplemented with 10% heat-inactivated calf bovine serum (Sigma-Aldrich), 10,000 U/mL penicillin, 10 μg/mL streptomycin, and 0.5 mg/mL neomycin, incubated overnight at 37 °C in a humidified incubator with 5% CO_2_. The next day the bacterial suspensions with various optical densities (OD_600nm_ = 0.01, 0.1, and 1.0) were added to reach a multiplicity of infection (MOI) ranging from 10 to 500 [45]. Plates were centrifuged at 100× *g* for 10 min at room temperature and incubated at 37 °C with 5% CO_2_ for 3 h. For the adhesion and invasion assay, separate parallel plates were used.

After incubation, plates for adhesion were washed three times with PBS followed by addition of 1 mL Triton X-100 (Calbiochem/Sigma-Aldrich) solution (0.1% v/v) to solubilize the INT 407 cells. To assess the total number of adhered and internalized bacteria, 10 μL aliquots of the suspended cells were plated on CCDA and incubated under microaerophilic conditions for 48 h at 42 °C.

The gentamicin protection assay (GPA) was used to quantify the number of internalized bacteria [59]. For invasion studies, after 3 PBS washes an additional incubation (60 min) with 1 mL RPMI with gentamicin (Aventis, Paris, France) at a bactericidal concentration (20 μg/mL) was performed to kill the attached but not internalized bacteria. After washing the plates three times with PBS, 1 mL Triton X-100 (0.1%) was added, and 10 μL volumes of the samples were plated as described above.

To assess the number of adhered bacteria, the number of invaded bacteria (counted from the gentamycin treated “invasion plates”) was subtracted from the total numbers of adhered and invaded bacteria (counted from the “adhesion plates”). The adherence and invasive capacities were expressed in percentages. Percentage adhesion values refers to the proportion of the total number of bacteria added to the epithelial cells that adhered. Percentage invasion refers to the proportion of the total number of adhered bacteria that were internalized [59].

### 2.5. Isolation of RNA from the Cultured and INT 407 Cell Invaded C. jejuni

A 3 h GPA invasion assay was carried out with *C. jejuni* strain CjTD-119 on the INT 407 cell line by the technique described above. The same strain (24 mL OD_600_ = 1) was used as a control in RPMI (without the presence of the INT 407) incubated at 37 °C in a humidified incubator with 5% CO_2_ for 3 h. RNAprotect Bacteria Reagent (Qiagen, Hamburg, Germany) was used to stabilize the RNA before collecting cells and trypsinization (Life Technologies, Carlsbad, California, USA) of the invaded cells. Bacteria were centrifuged at 8000 × *g* for 15 min. Collected cells were homogenized in RNAzol (Molecular Research Center, Cinncinatti, Ohio, USA) in a 1.5 mL microcentrifuge tube. The tubes were dropped three times into liquid nitrogen (-196 °C) for more effective extraction of the RNA from the intracellular bacteria. The total RNA concentration and purity was measured by an ND–1000 Spectrophotometer (Nanodrop, Thermo Scientific, Carlsbad, California, USA). The MICROBEnrich kit (Life Technologies) was used to remove the eukaryotic RNA in the sample.

### 2.6. Sequencing C. jejuni Strain CjTD-119

A library from the isolated chromosomal DNA was prepared with a Nextera XT (Illumina, USA) kit and sequenced on an Illumina MiSeq DNA sequencer using an Illumina V3 (600 cycles) kit. The average coverage was 861x. Assembly occurred with the MyPro pipeline [60] and the contigs were further assembled with Geneious 8 software (Geneious, Auckland, New Zealand). Validity of the assembled larger contigs were tested by remapping the reads. The accession number of the deposited genome is under the Bioproject PRJNA385384, with AMN06888245 NEWH00000000; Biosample SAMN06888245. For multilocus sequence typing (MLST), sequences of the seven housekeeping genes (*aspA*, *glnA*, *gltA*, *glyA*, *pgm*, *tkt*, and *uncA*) were uploaded to https://pubmlst.org/bigsdb?db=pubmlst_campylobacter_seqdef&page=sequenceQuery.

### 2.7. Whole Transcriptome Analysis (RNA-Seq)

The total gene expression profiles of intracellular and control bacteria after 3 h incubation was compared. RNA qualitative and quantitative measurements were implemented on Bioanalyzer (Agilent Technologies, Santa Clara, California, USA) and Qubit (Life Technologies). High quality (RIN > 8.5) total RNA samples from three biological replicates were pooled and processed using the SOLiD total RNA-Seq Kit (Life Technologies), according to the manufacturer’s instructions. Briefly, 3 µg of pooled RNA was *DNaseI* treated, and the ribosomal RNA was depleted using the RiboZero Prokaryotic rRNA Removal Kit (Illumina, San Diego, California, USA). The leftover was fragmented using RNase III, the 50–200 nt fraction size-selected sequencing adaptors ligated, and the templates were reverse transcribed using ArrayScript RT. The cDNA library was purified with the Qiagen MinElute PCR Purification Kit (Qiagen), and size-selected on a 6% TBE–Urea denaturing polyacrylamide gel. The 150–250 nt cDNA fraction was amplified using AmpliTaq polymerase and purified by AmPureXP Beads (Beckman Coulter, Indianapolis, Indiana, USA). The concentration of each library was determined using the SOLiD Library TaqMan Quantitation Kit (Life Technologies). Each library was clonally amplified on SOLiD P1 DNA Beads by emulsion PCR (ePCR). Emulsions were broken with butanol, and the ePCR beads enriched for template-positive beads by hybridization with magnetic enrichment beads. Template-enriched beads were extended at the 3′ end in the presence of terminal transferase and a 3′ bead linker. Beads with the clonally amplified DNA were deposited onto sequencing slides and sequenced on a SOLiD5500XL Instrument (Thermo Fisher, Carlsbad, California, USA) using the 50-base sequencing chemistry.

### 2.8. Bioinformatics

Bioinformatic analysis of the whole transcriptome sequencing was performed in colour space using the CLC Genomics Workbench (Qiagen). Raw sequencing data were trimmed by removal of low quality, short sequences so that only 45–50 nucleotide-long sequences were used in further analysis. Sequences were mapped onto their own (*C. jejuni* strain CjTD-119) genome sequence, using the default parameters. Results were manually cured to remove false positive hits, which showed highly skewed mapping of reads. Only genes detected with at least a 1.5-fold increase or decrease in transcription levels after normalization were considered for further analysis.

## 3. Results

### 3.1. The Presence of Putative Virulence Genes

PCR results of the distribution of the 14 putative virulence genes among the 192 *C. jejuni* isolates are summarized in Figure 1. All strains possessed the toxin gene *cdtB,* the adhesion gene *cadF*, and the flagellar system gene *flgE2.* Further flagellar genes (*flaB, flhB,* and *flgB*) involved in flagellar biosynthesis were present in 96%, 97%, and 99% of the isolates, respectively, although all isolates proved to be motile. *docA* and *docB*, with a role in colonization, were present in 90% of the isolates. The invasion-associated genes *iamA* and *ciaB* were detected in 99% and 87% of the isolates, respectively. The *virB11* gene was found in two isolates implicating that these strains carried the pVir plasmid, also proven by plasmid preparation (data not shown). The methyl-accepting chemotaxis protein coding gene, *docC*, was present in approximately half of the strains. A similar distribution was revealed in the case of *wlaN* and *cgtB* involved in LOS synthesis, 44% and 63%, respectively.

In order to identify isolates harbouring sialylated LOS, we tested for the presence of *cstII* and *cstIII*. Many isolates were *cstII*-positive (35%), and only 16% of the strains were *cstIII*-positive. Altogether 49% of isolates were negative for both, *cstII* and *cstIII*. Finally, the *hcp* gene encoding for the type-6 secretion system (T6SS) was present in 27% of the tested isolates.

### 3.2. Quantitation of the ECMP Binding Assay

*C. jejuni* isolates showed a high level of variability in ECMP binding capacity. Taking the binding values 0.1% as the threshold of positivity, 16% of the strains were found to adhere to each of the three ECMPs. Comparing the data (Appendix A), 5% of the strains were only able to bind collagen type IV, 2.5% only to fibronectin, and 4% only to laminin. There was also a distribution among the isolates in their preferred ECMP: 26% of the isolates bound collagen type IV at the highest level, while 24% of the strains preferentially bound to fibronectin and 21% laminin. Twenty nine percent of the isolates were unable to bind to any of the investigated ECMPs.

### 3.3. Adhesion and Invasion Abilities

Adhesion and invasion abilities varied considerably. From the distribution profile four classes of relative invasiveness were defined (Table 1): (1) high adhesion but low invasion (e.g., CjTD-6, CjTD-100 and CjTD-120); (2) low adhesion but high invasion potential (e.g., CjTD-49, CjTD-64, and CjTD-94); (3) high adhesion and high invasion potential (e.g., CjTD-35, CjTD-79, and CjTD-119); and (4) low adhesion and low invasion (e.g., CjTD-16, CjTD-172, and CjTD-182). Altogether 5.3% of the tested strains were able to adhere to INT 407 at a very high level, but they could not enter the cells in our experimental setup. About 25% of the isolates could neither adhere to nor invade INT 407 cells. A total of 3.7% of the strains were able to adhere to and invade INT 407 at a very high level. In Group 2, a high percentage (65.8%) of the adherent bacteria invaded the eukaryotic cells despite their relatively low adhesion capacity. *C. jejuni* 81-176 and NCTC 81116 were used as reference strains for the adhesion and invasion assays. Strain 81-176 is considered to have high invasion and adhesion ability [59]. The studied *C. jejuni* isolates were divided into four groups based on their abilities to adhere to and invade INT 407 human intestinal epithelial cells compared to reference strains 81-176 and NCTC 81116.

### 3.4. Sequence of C. jejuni Strain CjTD-119

Sequence information of strain CjTD-119 was constructed to 14 contigs. MLST analysis based on the seven housekeeping genes revealed that this isolate belongs to the clonal complex ST-42.

### 3.5. Bacterial Gene Expression during Invasion

Altogether 1668 open reading frames (ORFs) were detected with different fold changes and unique gene reads. At the third hour of invasion the expression of 1121 genes was significantly affected: 963 were upregulated and 158 were downregulated (Figure 2). ORFs that were pathogenically important and have the most significantly elevated genes are presented in Appendix A. Genes with significantly elevated expression during invasion were identified and categorized according to their functions. Each functional category was characterized by the dominance of genes with increased expression. The ratio of decrease in each category ranged from 4% to 43%.

Results showed that 50% of the affected genes were involved in general biosynthesis/metabolism and membrane functions, such as binding, transport, and translocation (Figure 2A). The expression of several transmembrane proteins, such as Sec proteins (e.g.,: SecY (5.655), SecE (10.713), and SecG (1.950)) [61], the Tol-dependent translocation system (TolB (1.749)) [62], and pore-forming channel protein (OmpA (3.622)) [63] was elevated (Appendix A). There was also elevated expression of factors that were recently speculated to be involved in adhesion (*Cj0268*, *Cj0090*, *Cj0091*, *Cj1136*, *Cj0286c*, *Cj0379c*, *jlpA*, *capA*, *cjaA*, *flpA*, *fbpA*, *cadF*, *peb2*, and *peb3)*. However, expression of adhesion protein coding genes *peb1* and *peb4* were not significantly altered.

There was increased expression (3.564x) of *omp50,* a hypothetical outer membrane porin. *Cme* genes (e.g., *cmeB* (1.567) and *cmeA (1.314)*) coding for a Cme efflux pump proteins were upregulated except for *cmeC* [64,65]. A transmembrane protein (AA01_00144/Cj0268c) required for adhesion and colonisation in vitro [66,67] was elevated (2.881). Among the adhesion proteins (Appendix A), the determinants of four lipoproteins (JlpA, CapA, CjaA, and FlpA) were elevated at this later stage of the invasion process. The two autotransporter proteins coded by *capA* (12.809) and *capB* (5.124) were elevated, but because of their low unique gene reads, their expression values are only indicative. The genes coding for fibronectin-binding adhesion proteins (*flpA* (2.909) and *fbpA* (2.160)) with possible roles in colonization [68] had elevated expression during the invasion. Two additional lipoprotein-coding genes were overexpressed (AA01_01303/Cj0090) (1.879) and AA01_01302/Cj0091 (2.132), probably playing a role in adhesion and repressed by the CmeR regulator [65]. Expression of the gene *cj0588* involved in adhesion was also elevated [69]. We detected the increased expression of *peb3* (4.861) and *peb2* (2.443), two genes coding for well-known adhesion proteins [70], and there was increased expression of *cadF* (1.535), coding a highly characterized adhesion protein [15]. In contrast, expression of genes coding for Peb1 [16] and Peb4 [17] proteins were not significantly elevated in the third hour of invasion.

All Mre-based bacterial cytoskeleton proteins responsible for the determination of bacterial cell shape, including *pbpC* (4.455), *mreC* (2.831), *pbpB* (2.661), *rodA* (2.242), and *mreB* (1.533) [71], increased in the internalized bacterium (Appendix A).

A number of genes coding for surface-associated saccharides were elevated in the third hour of invasion, including a group of capsular polysaccharide (CPS) genes (Appendix A). The genes *kpsM* and *kpsE* were overexpressed 4.2-fold and 1.9-fold, respectively. Expression of the gene *kpsT* (3.897), coding for transport of capsule, and *kpsC* (3.447), responsible for capsule modification, increased. Several LOS-related ORFs (Appendix A) were upregulated, including the biosynthetic genes *galE* (4.054), *waaF* (3.357), a glucosyltransferase (AA01_00523/*Cj1135*) (2.417), *gmhE* (1.549), and *lpxB* (lipid A biosynthesis) (3.718) [72].

Several components of multiple iron uptake systems (iron-uptake ABC transporter ATP-binding proteins, siderophore-mediated iron uptake system, ferrous iron transport proteins, and hemin uptake system proteins) were also elevated (Appendix A).

Genes belonging to the third major group (Figure 2A) are involved in nucleic acid metabolism. The high prevalence of this mRNA indicates more intensive intracellular activity of *C. jejuni* inside the eukaryotic cells during invasion than in normal culture conditions.

Hypothetical proteins were overexpressed in 138 of 1121 cases (Figure 2A, Appendix A), including a hypothetical sulfoxide reductase (AA01_00241/*cj0379c*)*,* a glycosyltransferase (*cj1136*), and a possible chemotaxis protein (AA01_00161/*cj0286c*) recently described in adhesion, invasion, and colonisation [66,73,74]. 

Expression of the most important gene groups and some of their representatives in the internalized *C. jejuni* are listed in Table 2. 

Invasion and colonisation-associated ORFs primarily comprised the “other” genes found (5 genes, 6%). The expression of two invasion-related genes (*cipA* and *ciaB*) increased 2.6-fold and 2.0-fold, respectively (Appendix A), and a recently discovered lipoprotein coding for AA01_01166*/*cj0497 was elevated (2.242) [74]. The two ATP-dependent protease-coding genes *lon* (3.057) and *clpP* (1.931), which are involved in invasion and colonization at higher temperatures, were also elevated [75] together with the serine protease coding *htrA* involved in adherence and invasion [76]. The bipartite energy taxis receptors coded by *cetA* (2.329) and *cetB* (2.350) were elevated [77]. A number of colonization-associated genes part of the Liv-system (*livJ* (4.207) and *livK* (4.639)), AA01_01544/cj0561c *(*3.052), *dnaJ* (3.705), *pldA* [78], AA01_00241/cj0379c (2.885) [73], and *docA* (1.770) [79] were upregulated (Appendix A).

Among the regulatory systems (Appendix A), the two-component regulatory (TCM) systems were elevated (*dccS* (2.638) and *racS* (1.733)) while the non TCM-system regulators showed a slight decrease (*cmeR* (-1.644) and *hspR* (-1.441)) [80].

Overall, the toxin genes (Appendix A) were downregulated while the chemotaxis genes (Appendix A) were upregulated.

An elevated oxidative stress response (Appendix A) was detected (e.g., *ahpC* (1.670) and *sodB* (1.532)) [81], although this value was not significant with *katA* (1.25). In contrast general stress-response genes such as *groES* and *groEL* showed decreased expression.

A number of genes involved in “protein synthesis/modification” (Appendix A) were elevated during the internalization of bacteria. Beside many elongation factors, members of the O-linked glycosylation (Appendix A) and the *N*-linked glycosylation (Appendix A) [22,24] systems became activated.

The number of up- (28/1121) and downregulated genes associated with the flagellar machinery were almost equal (Appendix A), with early flagellar gene products (class I) being overexpressed and class II and class III flagellar genes being under expressed (Appendix A).

## 4. Discussion

Despite its importance as a zoonotic agent and the availability of more than 1600 assembled genomes (NCBI genome database), there is an incomplete understanding of *C. jejuni* pathogenesis. The reason for that is when compared with other enteric bacterial pathogens, *C. jejuni* lacks many virulence and colonization determinants that are typically used by bacterial pathogens to infect hosts. Furthermore, *C. jejuni* has a broad repertoire of factors and pathways preferentially used to establish commensalism in any animal hosts and to promote diarrheal disease in the human population [55]. These flexible properties of *C. jejuni* are encoded in a relatively small genome (~2.2 MB) that further enhances the complexity of *C. jejuni*.

The ability of each of our 190 clinical isolates to spread on the BHI semisolid agar confirmed the pivotal role of flagella in *C. jejuni* infection [10]. Since loss of *flaB*, *flhB*, and *flgB* was reported to lead to a non-motile phenotype [39], the 96%, 97%, and 99% distribution of these genes (Figure 1) could be the result of a small nuclear polymorphism (SNP) in the primer [82,83] binding regions of the affected isolates.

Flagella represent a typical virulence factor with multiple roles during pathogenesis. Beside its role in motility [10] and cell invasion induction as a T3SS-like transport apparatus [39], the high rate of expression of the early class flagellar gene products (Appendix A) coding for the transmembrane complex of flagellar transport and stator proteins [11,42] (Supplementary Figure 2), indicated its transport function, also in the internalized bacterium cell.

The 100% prevalence of *cdtB* in the investigated isolates is in agreement with previous data showing that CDT genes were highly abundant (90%–100%) if the isolate was of human or animal origin [7,84,85,86]. We could not confirm that CDT was associated with local acute inflammation [87], but lack of correlation between its presence and the clinical symptoms does not support that this toxin has a decisive role in the clinical outcome. This is in full agreement with a recent study that showed that campylobacteriosis caused by CDT-negative strains was clinically indistinguishable from patients infected with an isolate that produced high levels of CDT [88]. In connection with CDT of *C. jejuni* it was also hypothesized that the presence of this toxin was associated with increased adherence and invasion to HeLa cells [85]. Our results disprove these findings, since aside from the 100% presence of gene *cdtB*, isolates showed discrepancies among their adhesive and invasive potentials.

Determining the link between putative virulence factors and the manifested clinical symptoms with *C. jejuni* remains difficult. High prevalence of *cadF* (100%), *iamA* (99%), *docA* (90%), and *ciaB* (87%) in our strain collection highlighted the potential importance of these genes in *C. jejuni* pathogenesis and survival. This was indicated in previous genetic studies examining their presence among poultry, cattle and human isolates [7,8,89,90]. *CdtABC, cadF, docA,* and *iamA* were detected in 98%–100%, 98%–100%, 94%, and 1.6%–92% of the human isolates, respectively [7,8,89,90]. The *hcp* gene, coding for the Type VI secretion system (T6SS), was recently suggested as a potential new player. It was reported to have a role in survival and pathogenesis of *Campylobacter* spp. [91]. Its relatively low (27%) prevalence, however, contradicts its crucial role in the pathogenic process, although its presence was higher (39%, 9/23) among bloody diarrhoea isolates compared to diarrhoea cases (23%, 33/140) (Appendix A). The *hcp* gene was also present in both registered clinical cases of stomach ache. Nevertheless, our results (27%) were in line with a previous study based on retail chicken isolates from Northern Ireland [92].

Controversial results were also obtained in recent studies when combinations of virulence factors were investigated, including *cdt* and *iam* [93]. Phenotypic tests (e.g., adhesion and invasion) are one possible method to elucidate potential roles of different factors or their combinations in *C. jejuni* pathogenesis. For that purpose, we performed a study with high strain numbers and both genetic and phenotypic analyses.

ECMP binding is one of the first steps of colonization of *C. jejuni* pathogenesis [25]. Binding of fibronectin, a major ECMP secreted by epithelial cells, and the subsequent signalling processes were demonstrated as an essential step of internalization and preferentially occurred at the basolateral cell surface [34]. Therefore, we investigated the binding potential of *C. jejuni* strains to other characteristic ECMPs in basal lamina, including laminin and type IV collagen. There was a lack of correlation between fibronectin adhesion potential [34,94] and adhesion onto INT 407 cells, confirming that the adhesion process of these 192 strains of *C. jejuni* is a multicomponent process involving several simultaneously occurring attachment mechanisms [58]. However, our results clearly suggest that other ECMPs, such as laminin and type IV collagen, could further facilitate the adhesion process either in at the basolateral invasion route or the transepithelial infection route, potentially causing bacteraemia.

An arsenal of genes (*jlpA*, *capA*, *cjaA*, *flpA*, *fbpA*, *cadF*, *cj0268*, *cj0090*, *cj0091*, *cj1136*, *cj0286c*, and *cj0379c*) have previously been confirmed or suggested to play a role in *C. jejuni* adhesion [14,15,16,17,19,20,68]. The large differences between the adhesion potentials of ECMPs and INT 407 cells was in harmony with recent findings focusing on the adhesion potentials of *C. jejuni* isolates from different origins to biotic and abiotic surfaces [95,96]. The diverse protein composition expressed on the surface of individual isolates could strongly influence adhesion potential and thereby affect the outcome of infections [58]. This was further complicated by the fact that *C. jejuni* host-cell interactions are strongly influenced by lectin–glycan interactions, mediated by LOS and glycoproteins [22,23,24,25]. Recent studies demonstrated the importance of sialylated LOS in invasion [97], an increased occurrence of bloody diarrhoea, and a longer duration of symptoms [98,99]. Our findings did not reveal any correlation between the presence of *cstII* and *cgtB* [28,100] and the invasive potential and clinical outcomes of the isolates as previously demonstrated [101]. However the high rate of expression of various synthases, transferases, and epimerases that take part in LOS synthesis (Appendix A) and *O*- and *N*-linked glycosylation (Appendix A) strongly suggest that these systems have supportive roles in maintaining intracellular *C. jejuni* survival.

Due to the abovementioned differences in putative virulence factors [55] among the *C. jejuni* isolates, it is not surprising that our adhesion and invasion capacity findings were in harmony with recent data [102] where adhesion and invasion potentials of 34 retail meat *C. jejuni* isolates ranged across a large scale.

Epithelial cell invasion is thought to be an essential step in *C. jejuni* infection [103,104] and correlates with the severity of clinical symptoms [32]. The ability of *C. jejuni* to invade cultured cells is strain dependent [30,105,106] and in vitro data indicates the presence of non-invasive but pathogenic *C. jejuni* isolates [107]. The two non-invasive strains, both of them isolated from simple diarrhoea in our strain collection (Appendix A), are under further study.

However intracellular survival is a typical feature of *C. jejuni* and remains one of the most intriguing aspects of its infection with several unknown molecular occurrences. For our transcriptomic study the non-polarized INT 407 epithelial cell line was used, as it is one of the most widely used model systems to investigate *C. jejuni*–host cell interactions and invasion [15,19,35,37,38].

The high expression of the abovementioned adhesion factors in the third hour of infection could reflect that these molecules somehow support the stabilisation of vacuole internalisation in the epithelial cell line INT 407. Our results support the dual (adhesion and invasion) role of these proteins during pathogenesis. In contrast, the insignificant elevation of Peb1 [16] and Peb4 [17] clearly indicate their role is limited to adhesion without affecting intracellular life.

Internalisation drives the bacterium into a novel and stressful environment [108]. Although there is little data available about the stress response evoked by invasion in *C. jejuni*, the general increase in the oxidative stress response genes (*katA, ahpC, sodB,* and *dps*), and decrease in general stress response genes (*groES*, *groEL*, and *dnaK*) strongly suggested that inside the *Campylobacter*-containing vacuoles (CCV), the bacterial cell encountered an oxidative environment. Furthermore, the molecular defensive mechanism of the host cells includes release of various reactive oxygen species, including superoxide, hydrogen peroxide, and halogenated oxygen molecules. Although CCV does not fuse with the lysosome [109], there is an activation of the oxidative stress response genes [110] contributing to radical inactivation.

Recent studies reported that during oxidative stress conditions, *C. jejuni* transformed its spiral shape to a coccoid one [81]. Although we did not evaluate the morphological state of the internalized *C. jejuni*, decreased expression of *pgp1* and *pgp2* [13,111], two genes contributing to the helical cell shape, suggested that the bacterium lost its curved shape in the vacuole. In the rod shape formation, there is increased expression of penicillin-binding protein genes (*pbp*) [71,112], supporting our findings as elevated expression levels of *pbpA*, *pbpB,* and *pbpC* in the internalized *C. jejuni* following 3 h of internalisation were detected (Appendix A). The increased expression levels of the Mre-based cytoskeletal and rod shape determining proteins (MreB, MreC and RodA) strengthen this hypothesis.

*C. jejuni* is unusual for an intestinal pathogen in its ability to coat its surface with a CPS [113], and although capsules are virulence factors for other pathogens, the role of CPS in *C. jejuni* disease has not been well defined [113,114]. This polysaccharide structure may be necessary at the beginning of the bacterial interaction with the mucus layer, but it may be followed by downregulation of CPS production, leading to exposure of bacterial adhesins [25,115]. Overexpression of some crucial genes linked to surface-associated saccharide production and transport (*kpsM* (4.240), *kpsE* (1.882), *kpsT* (3.897), and *kpsC* (3.447); Supplementary Table 2/D), and the fact that use of the intracellular route could be an important pathogenic feature of *C. jejuni*, suggested that CPS could support intracellular survival and therefore contribute to virulence. This could be one explanation why capsule mutants showed a modest reduction in invasion of intestinal epithelial cells in vitro [116,117,118].

Similar to CPS, the expression of LOS structures showed an increase in the internalized *C. jejuni* cell (Table 2). Aside from colonization, LOS effectively protects the bacterial cell against small molecules, including β-defensins, polymyxin [119], hydrophobic molecules [120], and phenolic compounds [121]. The increased expression of associated genes (Appendix A) indicates that the vacuolized *C. jejuni* cell protected itself by thickening its LOS, along with the CPS. This response to hostile environments is a typical feature of biofilm-forming bacteria.

Adaptation to the above-presented synthetic processes and to the novel environment surrounding the bacterial cell requires accelerated transcriptomic activity. Increased expression of several genes encoding regulatory and signal transduction proteins (Appendix A) and metabolism (Appendix A) indicated these processes. From our results it was evident that at least six regulators positively maintained the internalized state of *C. jejuni*. The upheaval of 90% of the metabolic genes indicated the production of novel proteins (Appendix A), with dominance of biosynthetic and metabolic (244/279), energy metabolic (29/35), and protein synthetic (51/53) genes (Figure 2B). The moderate but characteristic transcriptional increase of the subunits of the NADH-quinone oxidoreductase (Appendix A) supports this process by pumping protons accumulated from metabolic processes across the plasma membrane [122], increasing ATP synthesis.

An increase in membrane and transport protein expression strongly suggests that the bacterium did not separate itself from its environment as in case of the viable but non-culturable (VBNC) form [123], but actively interacted with it. The high expression of genes (*secY* (5.655), *secE* (10.713), and *secG* (1.950)) associated with the evolutionarily conserved heterotrimeric membrane channel SecYEG supports this hypothesis, as most proteins trafficking across or into the bacterial cytoplasmic membrane occur via the translocon system, formed by the three Sec proteins [61]. This process could be further supported by two partially characterized translocases like *tolB* (1.749) [62] and *ompA* (3.622) [63] (Appendix A). The elevated level of iron transporters (Appendix A) indicates that iron is also a limiting factor in the vacuole, similar to other pathogen–host interactions [124].

This study aimed to find correlations between the presence of (i) putative virulence genes, (ii) results of the phenotypic assay, and (iii) clinical symptoms by comparing the features of 192 *C. jejuni* human isolates. Despite the large sample size, we revealed no strong correlation between the genetic determinants of the investigated putative virulence factors, phenotypic characteristics, and clinical symptoms. Results of the subsequent transcriptomic analysis performed 3 h after invasion outlined the major characteristics of internalized state of *C. jejuni*: 1) elevation of the oxidative stress response genes in response to the oxidative condition in CVV; 2) a protective sheath formed around the internalized *C. jejuni* cell using several cell structures and mechanisms, including capsule, LOS, *N*- and *O*- glycosylation systems; and 3) the internalized state is not a static, but a dynamic one maintained by different translocases and the membrane-integrated part of the flagellar apparatus. Furthermore, several genes with hypothetical or unknown functions were active, and warrant further study.

## Figures and Tables

**Figure 1 microorganisms-08-00531-f001:**
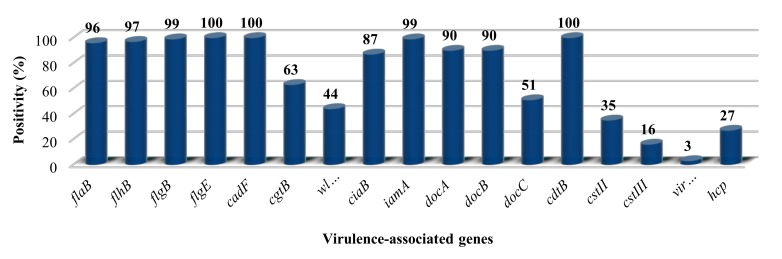
Percentage distribution of 14 previously described putative virulence genes in 190 *C. jejuni* isolates. (See details in Appendix A).

**Figure 2 microorganisms-08-00531-f002:**
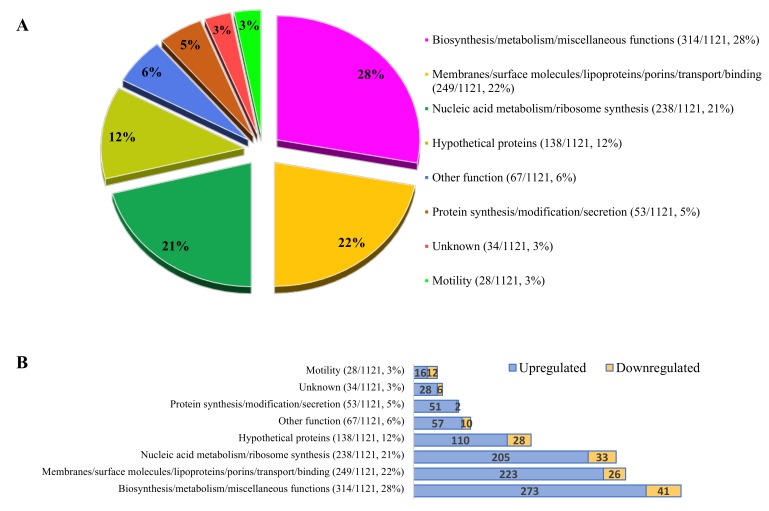
Distribution of the affected functions and genes in the third hour of invasion of the highly invasive strain CjTD-119. (**A**) Ratios of affected major functional groups (chart) and number of genes compared to the whole gene pool (in brackets). (**B**) Distribution of the up- and downregulated genes inside the major functional categories. Genes were assigned to the NCBI gene database. See Appendix A for more details.

**Table 1 microorganisms-08-00531-t001:** Adhesion/invasion capacity of the *C. jejuni* strains.

Group	Different Adhesion/Invasion Potential	% of Strains
Group 1	High adhesion/low invasion	5.3
Group 2	Low adhesion/high invasion	65.8
Group 3	High adhesion/high invasion	3.7
Group 4	Low adhesion/low invasion	25.2

**Table 2 microorganisms-08-00531-t002:** Summary of the representative genes with significantly increased transcriptional activities at the third hour of the invasion of strain CjTD-119.

Functional Group	Significantly Upregulated Genes
**Transmembrane Proteins**	
Sec protein system	*secE, secY, secG, secD, secF, yidC*
Tol-dependent translocation system	*tolB*
Pore forming channel proteins	*ompA, omp50*
Cme efflux pump proteins	*cmeAB*
Other transmembrane proteins	*cj0268c, lspA*
**Adhesion Proteins**	
Autotransporter proteins	*capA, capB*
Fibronectin binding adhesion proteins	*flpA, cj1349c*
Other adhesion proteins	*jlpA, cj0090, cj0091, cj0588, cadF*
**Bacterial Shape Determinant Genes**	
Mre-based bacterial cytoskeleton proteins	*pbpC, mreC, pbpB, rodA, mreB*
**Surface-Associated Saccharides**	
Capsular polysaccharide (CPS)	*kpsM, kpsT, kpsE, kpsC*
Lipooligosaccharide (LOS)	*galE*, *waaF*, *Cj1135*, *gmhE*, *lpxB*
**Invasion Proteins**	
ATP-dependent protease	*lon, clpP*
energy taxis receptors	*cetAB*
Other invasion proteins	*cipAB, htrA, cj0497*
**Iron Acquisition**	
Iron-uptake ABC transporter ATP-binding proteins	*cfbpABC*
The ferrous iron transport proteins	*feoAB*
Hemin uptake system proteins	*chuBCD*
Siderophore-mediated iron uptake system	*ceuCDE*
**Colonisation**	
Liv-system	*livJ, livK*
Other colonisation	*Cj0561c, dnaJ, pldA, Cj0379c, docA*
**Regulatory Systems**	
Two-component regulatory (TCM) systems	*dccS*
RacR–RacS system	*racS*
FlgSR system	*flgR*
Non TCM-system regulators	*spoT*
**Chemotaxis Genes**	*cheB*, *cheW*, *cheV, cheR, cheA*
**Flagellar Machinery**	
Stator proteins	*motAB*
Flagellar transport T3SS system	*flhAB, fliIPR*
Motor switch proteins	*fliY*
**Energy Metabolism**	
**Glycosylation System**	
“*O*”-linked glycosylation	*hisH, fabH2*
“*N*”-linked glycosylation	*pglAEFK, wlaDJ*
**Respiration**	*nuoACGIJKL*

**Protein Synthesis/Modification/Secretion**	*ssrA, tilS*

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
