# Peer review of "Virulence Traits of Inpatient Campylobacter jejuni Isolates, and a Transcriptomic Approach to Identify Potential Genes Maintaining Intracellular Survival"

_microorganisms, 2020, doi:10.3390/microorganisms8040531_

Round 1

Reviewer 1 Report

please refer to the attached comments.

Author Response

Please see the uploaded attachment.

Reviewer 2 Report

Manuscript ID: microorganisms-751933

Title: Virulence traits of inpatient Campylobacter jejuni isolates, and a transcriptomic approach to identify potential genes maintaining intracellular survival

Reviewer comments

Authors investigated virulence features of C. jejuni isolates from human patients to investigate the relationship between their virulence traits and clinical outcomes, performing also a transcriptomic analysis.

The presented results are very interesting and important, because they inform us about the unknown pathogenic mechanisms of C. jejuni, well known as a leading cause of bacterial foodborne gastroenteritidis worldwide. However, some revisions are necessary.

Abstract

I suggest rewriting the abstract by inserting an introduction, aim of study, material and methods, results and conclusions, in order to be clearer to readers

Line 14: I read 192 human C.jejuni strains but the total of hospitalized patients is 181. It is not clear to me.

Introduction

Line 81: please replace Campylobacter with Campylobacter

Materials and methods

Line 109: as reported in line 14, I do not understand where do these 192 strains come from. Do they include 190 C.jejuni plus the 2 reference strains? Please explain better this point.

Tables and Figures

In my opinion, tables and figures should be clearer.

Table 1: please order the 1st row

Figure 2: the imagines should be clearer. Please use colors or substitute with greyscale clearer for the readers

Table 2: please make the table more readable

Discussion: Although it is very interesting, it is very long. Please shorten it and comment more on the conclusions

Minor comments: please use italics for gene names along the entire manuscript.

Author Response

Please see uploaded attachment.
